# Continual Learning of a Mixed Sequence of Similar and Dissimilar Tasks

**Zixuan Ke**[1], **Bing Liu**[1,*] **and Xingchang Huang**[2]
[1] Department of Computer Science, University of Illinois at Chicago
[2] ETH Zurich
{zke4, liub}@uic.edu, huangxch3@gmail.com

## Abstract

Existing research on continual learning of a sequence of tasks focused on dealing with *catastrophic forgetting*, where the tasks are assumed to be dissimilar and have little shared knowledge. Some work has also been done to transfer previously learned knowledge to the new task when the tasks are similar and have shared knowledge. To the best of our knowledge, no technique has been proposed to learn a sequence of mixed similar and dissimilar tasks that can deal with forgetting and also transfer knowledge forward and backward. This paper proposes such a technique to learn both types of tasks in the same network. For dissimilar tasks, the algorithm focuses on dealing with forgetting, and for similar tasks, the algorithm focuses on selectively transferring the knowledge learned from some similar previous tasks to improve the new task learning. Additionally, the algorithm automatically detects whether a new task is similar to any previous tasks. Empirical evaluation using sequences of mixed tasks demonstrates the effectiveness of the proposed model.[2]

## 1 Introduction

In many applications, the system needs to incrementally or continually learn a sequence of tasks. This learning paradigm is called *continual learning* (CL) or *lifelong learning* (Chen and Liu, 2018). Ideally, in learning each new task $t$, the learner should **(1)** not forget what it has learned from previous tasks in order to achieve knowledge accumulation, **(2)** transfer the knowledge learned in the past forward to help learn the new task $t$ if $t$ is similar to some previous tasks and has shared knowledge with those previous tasks, **(3)** transfer knowledge backward to improve the models of similar previous tasks, and **(4)** learn a mixed sequence of dissimilar and similar tasks and achieve (1), (2) and (3) at the same time. To our knowledge, no existing CL technique has all these four capabilities. This paper makes an attempt to achieve all these objectives in the *task continual learning* (TCL) setting (also known as *task incremental learning*), where each task is a separate or distinct classification problem. This work generalizes the existing works on TCL. Note, there is also the *class continual learning* (CCL) setting (or *class incremental learning*), which learns a sequence of classes to build one overall multi-class classifier for all the classes seen so far.

As AI agents such as chatbots, intelligent personal assistants and physical robots are increasingly made to learn many skills or tasks, this work is becoming more and more important. In practice, when an agent faces a new task $t$, naturally some previous tasks are similar and some are dissimilar to $t$. The agent should learn the new task without forgetting knowledge learned from previous tasks while also improving its learning by transferring the shared knowledge from those similar tasks.

---

[*]Corresponding author. The work was done when B. Liu was at Peking University on leave of absence from University of Illinois at Chicago.
[2]https://github.com/ZixuanKe/CAT

Most existing CL models focus on (1), i.e., dealing with *catastrophic forgetting* or simply *forgetting* (Chen and Liu, 2018; Parisi et al., 2019; Li and Hoiem, 2016; Seff et al., 2017; Shin et al., 2017; Kirkpatrick et al., 2016; Rebuffi et al., 2017; Yoon et al., 2018; He and Jaeger, 2018; Yoon et al., 2018; Masse et al., 2018; Schwarz et al., 2018; Nguyen et al., 2018; Hu et al., 2019). In learning a new task, the learner has to update the network parameters but this update can cause the models for previous tasks to degrade or to be forgotten (McCloskey and Cohen, 1989). Existing works dealing with forgetting typically try to make the update of the network toward less harmful directions to protect the previously learned knowledge. Forgetting mainly affects the learning of a sequence of dissimilar tasks. When a sequence of similar tasks is learned, there is little forgetting as we will see in Section 4. There are also existing methods for knowledge transfer (Ruvolo and Eaton, 2013; Chen and Liu, 2014; Chen et al., 2015; Wang et al., 2019; Ke et al., 2020) when all tasks are similar.

This paper proposes a novel TCL model called **CAT** (*Continual learning with forgetting Avoidance and knowledge Transfer*) that can effectively learn a mixed sequence of similar and dissimilar tasks and achieve all the aforementioned objectives. CAT uses a *knowledge base* (**KB**) to keep the knowledge learned from all tasks so far and is shared by all tasks. Before learning each new task $t$, the learner first automatically identifies the previous tasks $T_{sim}$ that are similar to $t$. The rest of the tasks are dissimilar to $t$ and denoted by $T_{dis}$. In learning $t$, the learner uses the *task masks* (**TM**) learned from the previous tasks to protect the knowledge learned for those dissimilar tasks in $T_{dis}$ so that their important parameters are not affected (***no forgetting***). A set of masks (one for each layer) is also learned for $t$ in the process to be used in the future to protect its knowledge. For the set of similar tasks $T_{sim}$, the learner learns a *knowledge transfer attention* (**KTA**) to selectively transfer useful knowledge from the tasks in $T_{sim}$ to the new task to improve the new task learning (***forward knowledge transfer***). During training the new task $t$, CAT also allows the past knowledge to be updated so that some tasks in $T_{sim}$ may be improved as well (***backward knowledge transfer***). Our empirical evaluation shows that CAT outperforms the state of the art existing baseline models that can be applied to the proposed problem.

## 2 Related Work

Work on continual learning (CL) started in 1990s (see a review in (Li and Hoiem, 2016)). Most existing papers focus on dealing with *catastrophic forgetting* in neural networks. Li and Hoiem (2016) proposed the technique LwF to deal with forgetting using knowledge distillation. Kirkpatrick et al. (2016) proposed EWC to quantify the importance of network weights to previous tasks, and update weights that are not important for previous tasks. Similar methods are also used in (Aljundi et al., 2018; Schwarz et al., 2018; Zenke et al., 2017). Some methods memorize a small set of training examples in each task and use them in learning a new task to deal with forgetting (called *replay*) (Rebuffi et al., 2017; Lopez-Paz and Ranzato, 2017; Chaudhry et al., 2019; Wu et al., 2018; Kemker and Kanan, 2018). Some works built generators for previous tasks so that learning is done using a mixed set of real data of the new task and generated data of previous tasks (called *pseudo-replay*) (Shin et al., 2017; Kamra et al., 2017; Rostami et al., 2019; Hu et al., 2019).

Additionally, Rosenfeld and Tsotsos (2018) proposed to optimize loss on the new task with representations learned from old tasks. Hu et al. (2019) proposed PGMA, which deals with forgetting by adapting a shared model through parameter generation. Zeng et al. (2019) tried to learn the new task by revising the weights in the orthogonal direction of the old task data. Dhar et al. (2019) combined three loss functions to encourage the model resulted from the new task to be similar to the previous model. Other related works include Phantom Sampling (Venkatesan et al., 2017), Conceptor-Aided Backprop (He and Jaeger, 2018), Gating Networks (Masse et al., 2018; Serrà et al., 2018), PackNet (Mallya and Lazebnik, 2018), Diffusion-based Neuromodulation (Velez and Clune, 2017), IMM (Lee et al., 2017), Expandable Networks (Yoon et al., 2018; Li et al., 2019), RPSNet (Rajasegaran et al., 2019), reinforcement learning (Kaplanis et al., 2019; Rolnick et al., 2019), and meta-learning (Javed and White, 2019). See the surveys in (Chen and Liu, 2018; Parisi et al., 2019).

Most CL works focus on *class continual learning* (CCL). This paper focuses on *task continual learning* (TCL) (Fernando et al., 2017; Serrà et al., 2018). For example, GEM (Lopez-Paz and Ranzato, 2017) takes task id in addition to the training data of the specific task as input. A-GEM (Chaudhry et al., 2019) improves GEM's efficiency. HAT (Serrà et al., 2018) takes the same inputs and use hard attention to learn binary masks to protect old models in the TCL setting. But unlike CAT, HAT does not have mechanisms for knowledge transfer. Note that the controller in iTAML (Rajasegaran et al.,

2020) behaves similarly to HAT's annealing strategy in training. But the controller is for balancing between plasticity and stability, while HAT trains binary masks. UCL (Ahn et al., 2019) is a latest work on TCL. However, none of these methods can deal with forgetting and perform knowledge transfer to improve the new task learning at the same time. Progressive Network (Rusu et al., 2016) tries to perform forward knowledge transfer. It first builds one model for each task and then connects them. However, it cannot do backward transfer and its whole network size grows quadratically in the number of tasks, which makes it difficult to handle a large number of tasks. It also does not deal with a mixed sequence of tasks.

Earlier work on lifelong learning has focused on forward knowledge transfer to help learn the new task better (Thrun, 1998; Ruvolo and Eaton, 2013; Silver et al., 2013; Chen et al., 2015; Mitchell et al., 2015). Ke et al. (2020) and Wang et al. (2019) also did backward transfer. However, these lifelong learning works mainly use the traditional learning methods such as regression (Ruvolo and Eaton, 2013), naive Bayes (Chen et al., 2015; Wang et al., 2019), and KNN (Thrun, 1998) to build a model for each task independently and hence there is no forgetting problem. Although there are also works using neural networks (Ke et al., 2020; Wang et al., 2018; Thrun, 1998; Silver et al., 2013), all of them (including those based on traditional learning methods) work on tasks that are very similar and thus there is almost no forgetting. Earlier research on lifelong reinforcement learning worked on cross-domain lifelong reinforcement learning, where each domain has a sequence of similar tasks (Wilson et al., 2007; Bou Ammar et al., 2015). But like others, they don't deal with forgetting. To the best of our knowledge, no existing work has been done to learn a sequence of mixed similar and dissimilar tasks that deal with forgetting and improve learning at the same time.

## 3 Proposed CAT Model

The CAT model is depicted in Figure 1(A). At the bottom, it is the input data and the task ID $t$. Above it, we have the **knowledge base** (**KB**), which can be any differentiable layers (CAT has been experimented with a 2-layer fully connected network and a CNN architecture). The task ID $t$ is used to generate three different task ID embeddings. Two of them are element-wise multiplied ($\otimes$) with the outputs of the corresponding layers in the KB while the last one is the input to the knowledge transfer module (top right). The output of the KB can go to two branches (blue and red parallelogram). It will always go to the blue branch, which learns a classification model $f_{mask}$ for task $t$ and at the same time, learns a binary mask (called the **task mask** (**TM**)) for each layer in the KB indicating the units that are important/useful for the task in the layer. The mask is used to protect the learned knowledge for task $t$ in learning future tasks. If task $t$ is similar to some previous tasks, the output of the KB also goes to the right branch for selective knowledge transfer, which is achieved through **knowledge transfer attention** (**KTA**). On top of KTA, another classifier $f_{KTA}$ is built for task $t$, which leverages the transferred knowledge to learn $t$. This classifier should be used in testing rather than $f_{mask}$ for the task. $f_{mask}$ is only used in testing when the task $t$ has no similar previous task. Note that in task continual learning (TCL), task ID is needed because in testing each test instance will be tested by the classifier/model of its corresponding task (van de Ven and Tolias, 2019).[3] Note that Figure 1(A) does not show the network for detecting whether task $t$ is similar to some previous tasks, which we will discuss in Section 3.3. For now, we can assume that a set of similar previous tasks to $t$ is available.

### 3.1 Preventing Forgetting for Dissimilar Tasks: Task Masks

Let the set of tasks learned so far be $\mathcal{T}$ (before learning a new task $t$). Let $\mathcal{T}_{sim} \subseteq \mathcal{T}$ be a set of similar tasks to $t$ and $\mathcal{T}_{dis} = \mathcal{T} - \mathcal{T}_{sim}$ be the set of dissimilar tasks to $t$. We will discuss how to compute $\mathcal{T}_{sim}$ in Section 3.3. In learning $t$, we overcome forgetting for the dissimilar tasks in $\mathcal{T}_{dis}$ by identifying the units in the KB that are used by the tasks and blocking the gradient flow through the units (i.e., setting their gradients to 0). To achieve this goal, a task mask (a binary mask) $m_l^{(t)}$ is trained for each task $t$ at each layer $l$ of the KB during training for task $t$'s classifier/model, indicating which units are important for the task in the layer. Here we borrow the hard attention idea in (Serrà et al., 2018) and leverage the task ID embedding to train the mask.

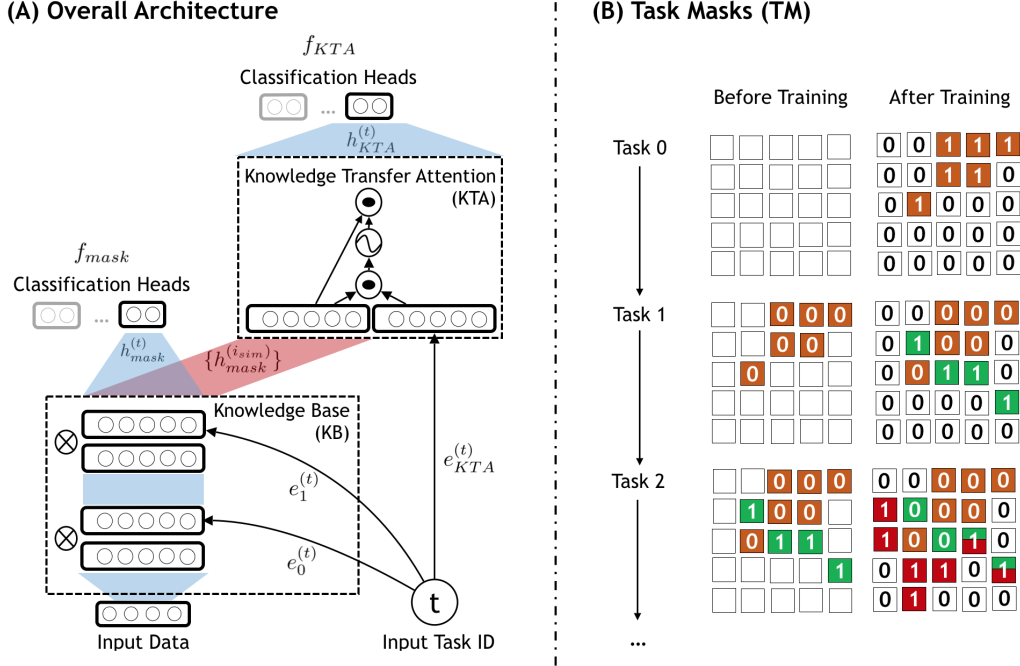

Figure 1: (A) CAT architecture, and (B) illustration of task masking and knowledge transfer. Some notes about (B) are: In the matrix before training, those cells with 0's are the units to be protected (masked), and with 1's are units to be shared and whose tasks are similar to the current task. Those cells without a number are free units (not used). In the matrix after training, those cells with 1's show those units that are important for the current task, which are used as a mask for the future. The rest of the cells or units are not important for the task. Those 0 cells without a color are not used by any task.

**From Task ID Embedding to Task Mask.** For a task ID $t$, its embedding $e_l^{(t)}$ consists of differentiable deterministic parameters that can be learned together with other parts of the network. The subscript $l$ indicates the layer number. A separate task ID embedding is trained for each layer of the KB. To generate the task mask $m_l^{(t)}$ from $e_l^{(t)}$, Sigmoid is used as a pseudo-gate function and a positive scaling hyper-parameter $s$ is applied to help training. $m_l^{(t)}$ is computed as follows:

$$m_l^{(t)} = \sigma(se_l^{(t)}) \tag{1}$$

Given the output of each layer in the KB, $h_l^{(t)}$, we element-wise multiply $h_l^{(t)} \otimes = m_l^{(t)}$. The masked output of the last layer $h_{mask}^{(t)}$ is fed to the $f_{mask}$ classification head to train the task classifier. After learning task $t$, the final $m_l^{(t)}$ is saved and added to the set $\{m_l^{(t)}\}$.

**Block Gradients Flow through Used Units for Dissimilar Tasks.** For each previous dissimilar task $i_{dis}$ in $\mathcal{T}_{dis}$ of the current task $t$, its mask $m_l^{(i_{dis})}$ indicates which units in the KB are used by task $i_{dis}$. In learning task $t$, $m_l^{(i_{dis})}$ is used to set the gradient $g_l^{(t)}$ on *all* used units of the layer $l$ to 0. Before modifying the gradient, we first accumulate all used units by all previous dissimilar tasks making use of their masks. Since $m_l^{(i_{dis})}$ is binary, we can use element-wise maximum to achieve the accumulation:

$$m_l^{(t_{ac})} = ElementMax(\{m_l^{(i_{dis})}\}) \tag{2}$$

$m_l^{(t_{ac})}$ is applied to the gradient:

$$g_l^{(t)} \otimes = (1 - m_l^{(t_{ac})}) \tag{3}$$

Those gradients corresponding to the 1 entries in $m_l^{(t_{ac})}$ are set to 0 while the others remain unchanged. Note that we expand (copy) the vector $m_l^{(t_{ac})}$ to match the dimensions of $g_l^{(t)}$.

**Training Tricks.** Though the idea is intuitive, $e_l^{(t)}$ is not easy to train. To make the learning of $e_l^{(t)}$ easier, an annealing strategy is applied (Serrà et al., 2018). That is, $s$ is annealed during training, inducing a gradient flow and set $s = s_{max}$ during testing. Eq. 1 approximates a unit step function as the mask, with $m_l^{(t)} \to \{0, 1\}$ when $s \to \infty$. A training epoch starts with all units being equally active, which are progressively polarized within the epoch. Specifically, $s$ is annealed as follows:

$$s = \frac{1}{s_{max}} + (s_{max} - \frac{1}{s_{max}})\frac{b-1}{B-1}, \tag{4}$$

where $b$ is the batch index and $B$ is the total number of batches in an epoch. The task masks are trained together with $f_{mask}$ by minimizing (using cross entropy):

$$\frac{1}{N_t}\sum_{i=1}^{N_t}\mathcal{L}(f_{mask}(x_i^{(t)};\theta_{mask}), y_i^{(t)}) \tag{5}$$

**Illustration.** In Figure 1(B). Task 0 is the first task. After learning it, we obtain its useful units marked in orange with a 1 in each unit, which serves as a mask for future tasks. In learning task 1, we found that task 1 is not similar to task 0. Those useful units for task 0 is masked (with 0 in those orange units or cells in the matrix on the left). The process also learns the useful units for task 2 marked in green with 1's.

## 3.2 Knowledge Transfer from Similar Tasks: Knowledge Transfer Attention

If there are similar tasks ($\mathcal{T}_{sim} \neq \emptyset$) to the new task $t$, we want to learn $t$ better, by encouraging knowledge transfer from $\mathcal{T}_{sim}$. Due to the fact that every task may have its domain specific knowledge that is not applicable to other tasks, knowledge transfer has to be selective. We propose a *knowledge transfer attention* (KTA) for the purpose. The idea is to give different importance to different previous tasks by an attention mechanism so that their knowledge can be selectively transferred to the new task. Those transferred units are also made updateable to achieve *backward knowledge transfer* automatically so that the previous task models may be improved in training the new task $t$.

**Knowledge Base Output for Previous Tasks.** Recall we know which units are for which task $j$ by reading $\{m_l^{(j)}\}$. For each previous similar task $i_{sim}$ in $\mathcal{T}_{sim}$ of the current task $t$, we can compute its masked KB output $h_{mask}^{(i_{sim})}$ (the last layer) by applying $m_l^{(i_{sim})}$ to the KB.

**Knowledge Transfer Attention.** We learn another task ID embedding (separate from the task ID embedding used in the KB) $e_{KTA}^{(t)}$ and stack all the outputs of the last layer in the KB $h_{mask}^{(i_{sim})}$ into a collection $\{h_{mask}^{(i_{sim})}\}$. We then compute the attention weight for each $h_{mask}^{(i_{sim})}$ in the collection in the right branch of Figure 1(A) by:

$$a^{(i_{sim})} = softmax(\frac{(e_{KTA}^{(t)}\theta_q)(\{h_{mask}^{(i_{sim})}\}\theta_k)^\intercal}{\sqrt{d_k}}) \tag{6}$$

where $d_k$ is the number of previous similar tasks ($|\mathcal{T}_{sim}|$). $\theta_q$ and $\theta_k$ are parameters matrices for projections in self-attention (Vaswani et al. (2017)). The $a^{(i_{sim})}$ indicates the importance of each previous task in $\{h_{mask}^{(i_{sim})}\}$. We then compute the weighted sum among $\{h_{mask}^{(i_{sim})}\}$ to get the output of similar tasks:

$$h_{KTA}^{(t)} = \sum_i a^{(i_{sim})}(\{h_{mask}^{(i_{sim})}\}\theta_v) \tag{7}$$

where $\theta_v$ is the parameter matrix for projection in self-attention. $h_{KTA}^{(t)}$ is then fed to the $f_{KTA}$ classification head to learn the classifier.

**Loss function for training $f_{mask}$ and $f_{KTA}$.** Both terms below use cross entropy.

$$\frac{1}{N_t}\sum_{j=1}^{N_t}\mathcal{L}(f_{mask}(x_j^{(t)};\theta_{mask}), y_j^{(t)}) + \frac{1}{N_t}\sum_{j=1}^{N_t}\mathcal{L}(f_{KTA}(x_j^{(t)};\theta_{KTA}), y_j^{(t)}), \tag{8}$$

where $N_t$ is the number of training examples in task $t$, $\theta_{mask}$ is the set of parameters of $f_{mask}$, and $\theta_{KTA}$ is the set of parameters of $f_{KTA}$.

**Illustration.** In Figure 1(B), when task 2 arrives, the system found that task 1 is similar to task 2, but task 0 is not. Then, task 0's important units are blocked, i.e., its mask entries are set to 0 (orange units in the left matrix). Task 1's important units are left open with its mask entries set to 1 (green units in the left matrix). After learning with knowledge transfer, task 2 and task 1 have some shared units that are important to both of them, i.e., those units marked in both red and green.

### 3.3 Task Similarity Detection

In the above discussion, we assume that we know the set of similar previous tasks $\mathcal{T}_{sim}$ of the new task $t$. We now present how to find similar tasks for a given task $t$. We use a binary vector TSV (*task similarity vector*) to indicate whether each previous task $k \in \mathcal{T}$ is similar to the new task $t$.

We define task similarity by determining whether there is a *positive knowledge transfer* from a previous task $k$ to the current task $t$. A *transfer model* $f_{k \to t}$ is used to transfer knowledge from task $k$ to task $t$. A single task model $f_\varnothing$, called the *reference model*, is used to learn $t$ independently. If the following statistical risk holds, which indicates a positive knowledge transfer, we say that task $k$ is similar to task $t$; otherwise task $k$ is dissimilar to task $t$.

$$\underset{(\mathcal{X}^{(t)}, \mathcal{Y}^{(t)})}{\mathbb{E}} [\mathcal{L}(f_{k \to t}(\mathcal{X}^{(t)}; \theta_{k \to t}), \mathcal{Y}^t)] > \underset{(\mathcal{X}^{(t)}, \mathcal{Y}^{(t)})}{\mathbb{E}} [\mathcal{L}(f_\varnothing(\mathcal{X}^{(t)}; \theta_\varnothing), \mathcal{Y}^{(t)})] \quad (9)$$

We use a validation set to check whether Eq. 9 holds. Specifically, if the transfer model $f_{k \to t}$ classifies the validation data of task $t$ better than the reference model $f_\varnothing$, then we say $k$ contains shareable prior knowledge that can help $t$ learn a better model than without the knowledge, $f_\varnothing$, indicating positive knowledge transfer. We set $TSV^{(t)}[k] = 1$ indicating that $k$ is similar to $t$; otherwise $TSV^{(t)}[k] = 0$ indicating that $k$ is dissimilar to $t$.

**Transfer model.** The transfer model $f_{k \to t}$ trains a small readout function (1 layer fully-connected network on top of the KB) for the previous task $k$ given the representation/features for $x^{(t)}$ produced by the task $k$ model in the KB. In training the transfer network, the KB is frozen or not updated.

Recall the model of task $k$ is specified by its mask $\{m_l^{(k)}\}$, which is saved after training task $k$. The transfer network is trained by minimizing the following empirical risk using the cross entropy loss:

$$\frac{1}{N_t} \sum_{i=1}^{N_t} \mathcal{L}(f_{k \to t}(x_i^{(t)}; \theta_{k \to t}), y_i^{(t)}) \quad (10)$$

**Reference model.** The reference model $f_\varnothing$ is a separate network for building a model for task $t$ alone from scratch with random initialization. It uses the same architecture as $f_{k \to t}$ without applying any task masks. However, the size of the network is smaller, 50% of $f_{k \to t}$ in our experiments. The network is trained by the following using the cross entropy loss:

$$\frac{1}{N_t} \sum_{i=1}^{N_t} \mathcal{L}(f_\varnothing(x_i^{(t)}; \theta_\varnothing), y_i^{(t)}) \quad (11)$$

## 4 Experiments

We now evaluate CAT following the standard continual learning evaluation method in (Lange et al., 2019). We present CAT a sequence of tasks for it to learn. Once a task is learned, its training data is discarded. After all tasks are learned, we test all task models using their respective test data. In training each task, we use its validation set to decide when to stop training.

### 4.1 Experiment Datasets

Since CAT not only aims to deal with forgetting for dissimilar tasks but also to perform knowledge transfer for similar tasks, we consider two types of datasets.

*Similar Task Datasets:* We adopt two similar-task datasets from *federated learning*. Federated learning is an emerging machine learning paradigm with its emphasis on data privacy. The idea is to train through model aggregation rather than the conventional data aggregation and keep local data staying on the local device. These datasets naturally consist of similar tasks. We randomly choose 10 tasks

| Dataset | # Classes | # Training | # Validation | # Testing |
|---|---|---|---|---|
| M(EMNIST-10/20, F-EMNIST) | | | | |
| EMNIST-10/20 | 47 | 6,000 | 6,000 | 800 |
| F-EMNIST | 62 | 6,911 | 762 | 858 |
| M(CIFAR100-10/20, F-CelebA) | | | | |
| CIFAR100-10/20 | 100 | 45,000 | 5,000 | 10,000 |
| F-CelebA | 2 | 870 | 90 | 110 |

Table 1: Statistics of the datasets, which contain the total number of classes and the total numbers of training, validation and testing instances for each dataset.

from two publicly available federated learning datasets (Caldas et al., 2018) to form (1) **F-EMNIST** - each of the 10 tasks contains one writer's written digits/characters (62 classes in total), and (2) **F-CelebA** - each of the 10 tasks contains images of a celebrity labeled by whether he/she is smiling or not. Note that the training and testing sets are already provided in (Caldas et al., 2018). We further split about 10% of the original training data as the validate data.

*Dissimilar Task Datasets:* We use 2 benchmark image classification datasets: **EMNIST** (LeCun et al., 1998) and **CIFAR100** (Krizhevsky et al., 2009). We consider two splitting scenarios. For each dataset, we prepare two sets of tasks with a different number of classes in each task. For EMNIST, which has 47 classes in total, the first set has 10 tasks and each task has 5 classes (the last task has 2 classes). The second set has 20 tasks and each task has 2 classes (the last task has 9 classes). For CIFAR100, which has 100 classes, the first set has 10 tasks and each task has 10 classes. The second set has 20 tasks and each task has 5 classes. For EMNIST, for efficiency reasons we randomly sampled a subset of the original dataset and also make the validation set the same as the training set following (Serrà et al., 2018). For CIFAR100, we split 10% of the training set and keep it for validation purposes. Statistics of the datasets are given in Table 1.

*Mixed Sequence Datasets for Experimentation:* To experiment with learning a mixed sequence of tasks, we constructed four *mixed sequence datasets* from the above similar and dissimilar tasks datasets: **M(EMNIST-10, F-EMNIST)**, **M(EMNIST-20, F-EMNIST)**, **M(CIFAR100-10, F-CelebA)**, and **M(CIFAR100-20, F-CelebA)**. The first mixed sequence dataset **M(EMNIST-10, F-EMNIST)** consists of 5 random sequences of tasks from EMNIST (10 dissimilar tasks) and all 10 similar F-EMNIST task. **M(EMNIST-20, F-EMNIST)** consists of 5 random sequences of tasks from EMNIST (20 dissimilar tasks) and all 10 similar F-EMNIST tasks. Note that EMNIST and F-EMNIST datasets are paired together because they contain images of the same size. The other two mixed sequence datasets involving CIFAR100 and F-CelebA are prepared similarly. CIFAR100 and F-CelebA are paired together also because their images are of the same size.

## 4.2 Compared Baselines

We consider ten task continual learning (TCL) baselines. **EWC** (Kirkpatrick et al., 2016) - a popular regularization-based class continual learning (CCL) method. We adopt its TCL variant implemented by Serrà et al. (2018). **HAT** (Serrà et al., 2018) - one of the best TCL methods with almost no forgetting. **UCL** (Ahn et al., 2019) - a latest TCL method using a Bayesian online learning framework. **HYP** (von Oswald et al., 2020) - a latest TCL method addressing forgetting by generating the weights of the target model based on the task ID. **HYP-R** (von Oswald et al., 2020) - a latest replay-based method. **PRO** (Progressive Network) (Rusu et al., 2016) - another popular continual learning method which focuses on forward transfer. **PathNet** (Fernando et al., 2017) - a classical continual learning method selectively masks out irrelevant model parameters. We also adopt its TCL variant implemented by Serrà et al. (2018). **RPSNet** (Rajasegaran et al., 2019) - an improvement over PathNet which encourages knowledge sharing and reuse. As RPSNet is a CCL method, we adapted it to a TCL method. Specifically, we only train on the corresponding head of the specific task ID during training and only consider the corresponding head's prediction during testing. **NCL (naive continual learning)** - greedily training a sequence of tasks incrementally without dealing with forgetting. It uses the same network as the next baseline. **ONE (one task learning)** - building a model for each task independently using a separate neural network, which clearly has no knowledge transfer and no forgetting involved.

| | NCL | ONE | EWC | UCL | HYP | HYP-R | PRO | PathNet | RPSNet | HAT | **CAT** |
|---|---|---|---|---|---|---|---|---|---|---|---|
| M(EMNIST-10, F-EMNIST): Overall | 0.7293 | 0.7337 | 0.7339 | 0.7262 | 0.6271 | 0.4889 | 0.5391 | 0.5901 | 0.7044 | 0.7302 | **0.7710** |
| M(EMNIST-10, F-EMNIST): EMNIST-10 | 0.9156 | **0.9437** | 0.9157 | 0.9161 | 0.8329 | 0.7254 | 0.9289 | 0.9163 | 0.8945 | 0.9337 | 0.9287 |
| M(EMNIST-10, F-EMNIST): F-EMNIST | 0.5430 | 0.5238 | 0.5521 | 0.5362 | 0.4212 | 0.2524 | 0.1492 | 0.2638 | 0.5144 | 0.5268 | **0.6134** |
| M(CIFAR100-10, F-CelebA): Overall | 0.5535 | 0.5967 | 0.5945 | 0.5523 | 0.5352 | 0.3703 | 0.5863 | 0.5504 | 0.4801 | 0.5682 | **0.6194** |
| M(CIFAR100-10, F-CelebA): CIFAR100-10 | 0.5124 | **0.5861** | 0.5345 | 0.5373 | 0.4667 | 0.2096 | 0.5599 | 0.5244 | 0.4056 | 0.5692 | 0.5479 |
| M(CIFAR100-10, F-CelebA): F-CelebA | 0.5945 | 0.6073 | 0.6545 | 0.5673 | 0.6036 | 0.5309 | 0.6127 | 0.5764 | 0.5545 | 0.5673 | **0.6909** |
| M(EMNIST-20, F-EMNIST): Overall | 0.8024 | 0.8245 | 0.8213 | 0.8186 | 0.7332 | 0.6092 | 0.6794 | 0.7115 | 0.74835 | 0.8169 | **0.8439** |
| M(EMNIST-20, F-EMNIST): EMNIST-20 | 0.9270 | **0.9712** | 0.9393 | 0.9567 | 0.8970 | 0.7856 | 0.9660 | 0.9472 | 0.8861 | 0.9678 | 0.9566 |
| M(EMNIST-20, F-EMNIST): F-EMNIST | 0.5531 | 0.5310 | 0.5855 | 0.5425 | 0.4056 | 0.2565 | 0.1062 | 0.2403 | 0.4728 | 0.5136 | **0.6187** |
| M(CIFAR100-20, F-CelebA): Overall | 0.6018 | 0.6796 | 0.6292 | 0.6368 | 0.5878 | 0.3892 | 0.6682 | 0.6169 | 0.5410 | 0.6535 | **0.6843** |
| M(CIFAR100-20, F-CelebA): CIFAR100-20 | 0.6136 | **0.7058** | 0.6348 | 0.6689 | 0.6053 | 0.3274 | 0.6896 | 0.6163 | 0.5507 | 0.6802 | 0.6683 |
| M(CIFAR100-20, F-CelebA): F-CelebA | 0.5782 | 0.6273 | 0.6182 | 0.5727 | 0.5527 | 0.5127 | 0.6255 | 0.6182 | 0.5218 | 0.6000 | **0.7164** |

Table 2: Accuracy results of different models on the four mixed sequence datasets (average over 5 random sequences) using a 2-layer fully connected network. The number in bold in each row is the best result of the row.

| | NCL | ONE | HAT | **CAT** |
|---|---|---|---|---|
| M(CIFAR100-10,F-CelebA): Overall | 0.6155 | 0.6764 | 0.6178 | **0.6831** |
| M(CIFAR100-10,F-CelebA): CIFAR100-10 | 0.5692 | **0.6892** | 0.6301 | 0.6099 |
| M(CIFAR100-10,F-CelebA): F-CelebA | 0.6618 | 0.6636 | 0.6055 | **0.7564** |
| M(CIFAR100-20,F-CelebA): Overall | 0.6931 | 0.7428 | 0.6946 | **0.7468** |
| M(CIFAR100-20,F-CelebA): CIFAR100-20 | 0.6669 | **0.7870** | 0.7419 | 0.7339 |
| M(CIFAR100-20,F-CelebA): F-CelebA | 0.7455 | 0.6545 | 0.6000 | **0.7727** |

Table 3: Accuracy results of different models on the mixed sequences of F-CelebA and CIFAR100 (average over 5 random sequences) using an AlexNet-like architecture. The number in bold in each row is the best result of the row.

## 4.3 Network and Training Details

Unless otherwise stated, for NCL, ONE, and the KB in CAT, we employ a 2-layer fully connected network for all our datasets. For F-CelebA and CIFAR100, we further experiment CAT using a CNN based AlexNet-like architecture (Krizhevsky et al., 2012). We also employ the embedding with 2000 dimensions as the final and hidden layer of the KB. The task ID embeddings have 2000 dimensions. A fully connected layer with softmax output is used as the $f_{mask}$ and $f_{KTA}$ classification heads, together with the categorical cross-entropy loss. We use 140 for $s_{max}$ in $s$, dropout of 0.5 between fully connected layers. For the knowledge transfer attention (KTA), we apply layer normalization and dropout following the setting in (Vaswani et al., 2017). We also employ multiple attention heads. We set the number of attention heads to 5 (grid search from the candidates set $\{1, 5, 10, 15, 20\}$ on the validation set). We train all models using SGD with the learning rate of 0.05. We stop training when there is no improvement in the validation accuracy for 5 consecutive epochs (i.e., early stopping with $patience = 5$). The batch size is set to 64. For all the other baselines, we use the code provided by their authors and adopt their original parameters.

## 4.4 Results and Analysis

Table 2 gives the accuracy results of all systems on the four mixed sequence datasets. For NCL, ONE, HAT, and CAT, we use a 2-layer fully connected network. The first part of the table contains the average results for the 5 random sequences of the 20 tasks M(EMNIST-10, F-EMNIST) dataset. The first row shows the average accuracy of all 20 tasks for each system. The second row shows the average accuracy results of only the 10 dissimilar tasks of EMNIST-10 among all 20 tasks of the compared systems. The third row shows the average accuracy results of only the 10 similar tasks of F-EMNIST among all 20 tasks of the compared systems. Other parts of the table contain the corresponding results of the 20/30 tasks mixture sequence datasets. The detailed results of different sequences are given in *Supplementary Materials*. Table 3 gives the results of the two mixed sequence datasets involving CIFAR100 and F-CelebA using the CNN based AlexNet-like architecture.

| Task | ONE | Backward | Forward |
|---|---|---|---|
| M(EMNIST-10, F-EMNIST): F-EMNIST | 0.5238 | **0.6134** | 0.6104 |
| M(EMNIST-20, F-EMNIST): F-EMNIST | 0.5310 | **0.6187** | 0.6081 |
| M(CIFAR100-10, F-CelebA): F-CelebA | 0.6073 | **0.6909** | 0.6873 |
| M(CIFAR100-20, F-CelebA): F-CelebA | 0.6273 | **0.7164** | 0.6782 |

Table 4: Effect of forward and backward knowledge transfer in CAT.

|  | CAT (-TSV: all-sim; -KTA) | CAT (-TSV: all-sim) | CAT (-TSV: all-dis) | CAT (-KTA) | **CAT** |
|---|---|---|---|---|---|
| M(EMNIST-10, F-EMNIST): Overall | 0.7293 | 0.1887 | 0.7337 | 0.7585 | **0.7710** |
| M(EMNIST-10, F-EMNIST): EMNIST-10 | 0.9156 | 0.2961 | **0.9437** | 0.9301 | 0.9287 |
| M(EMNIST-10, F-EMNIST): F-EMNIST | 0.5430 | 0.0813 | 0.5238 | 0.5870 | **0.6134** |
| M(CIFAR100-10, F-CelebA): Overall | 0.5535 | 0.3090 | 0.5967 | 0.5915 | **0.6194** |
| M(CIFAR100-10, F-CelebA): CIFAR100-10 | 0.5124 | 0.1253 | **0.5861** | 0.5594 | 0.5479 |
| M(CIFAR100-10, F-CelebA): F-CelebA | 0.5945 | 0.4927 | 0.6073 | 0.6236 | **0.6909** |
| M(EMNIST-20, F-EMNIST): Overall | 0.8024 | 0.3631 | 0.8245 | 0.8423 | **0.8439** |
| M(EMNIST-20, F-EMNIST): EMNIST-20 | 0.9270 | 0.5174 | **0.9712** | 0.9637 | 0.9566 |
| M(EMNIST-20, F-EMNIST): F-EMNIST | 0.5531 | 0.0545 | 0.5310 | 0.5995 | **0.6187** |
| M(CIFAR100-20, F-CelebA): Overall | 0.6018 | 0.3207 | 0.6796 | 0.6572 | **0.6843** |
| M(CIFAR100-20, F-CelebA): CIFAR100-20 | 0.6136 | 0.2210 | **0.7058** | 0.6740 | 0.6683 |
| M(CIFAR100-20, F-CelebA): F-CelebA | 0.5782 | 0.5200 | 0.6273 | 0.6236 | **0.7164** |

Table 5: Ablation experiment results.

**Overall Performance.** The *overall* accuracy results of all tasks for four mixed sequence datasets in Tables 2 and 3 show that CAT outperforms all baselines. In Table 2, although EWC, HAT, and UCL perform better than NCL due to their mechanisms for avoiding forgetting, they are all significantly worse than CAT as they don't have methods to encourage knowledge transfer. HYP, HYP-R, PathNet and RPSNet fail to outperform NCL, indicating their failure in learning mixed sequence of tasks. Even though PRO, by construction, never forget, it still exhibits difficulties in dealing with mixed sequences.

**Performance on Dissimilar Tasks.** We can see from Tables 2 and 3 that CAT performs better than most of the baselines on dissimilar tasks. It is not surprising that ONE is the best overall except for EMNIST-20 as it builds separate classifiers independently. CAT performs similarly to HAT, which has little forgetting. This indicates that CAT deals with forgetting reasonably well. In Table 2, we see that PathNet and RPSNet work well on EMNIST-20 datasets but extremely poorly on F-EMNIST, indicating they are not able to deal with a mixed sequence well as CAT does.

**Performance on Similar Tasks.** For the results of similar tasks, In both Tables 2 and 3, we can see that CAT markedly outperforms HAT and all other baselines as CAT can leverage the shared knowledge among similar tasks while the other CL approaches, including HAT, only tries to avoid interference with the important units of previous tasks to deal with forgetting.

**Effectiveness of Knowledge Transfer.** Here, we look at only the similar tasks from F-EMNIST and F-CelebA in the four mixed sequence experiments. For forward transfer, we use the test accuracy of each similar task in F-EMNIST or F-CelebA when it was first learned. For backward transfer, we use the final result after all tasks are learned. The average results are given in Table 4. Table 4 clearly shows that forward knowledge transfer is highly effective. For backward transfer, CAT slightly improves the forward performance for F-MNIST and markedly improves the performance for F-CelebA.

### 4.5 Ablation Experiments

We now show the results of ablation experiments in Table 5. "-KTA" means without deploying KTA (knowledge transfer attention). "-TSV" means without detecting task similarity, i.e., no TSV (task similarity vector), which has two cases, treating all previous tasks as dissimilar (all-dis) or treating all previous tasks as similar (all-sim). From Table 5, we can see that the full CAT system always gives the best overall accuracy and every component contributes to the model.

## 5 Conclusion

This paper first described four desired capabilities of a continual learning system: no forgetting, forward knowledge transfer, backward knowledge transfer, and learning a mixed sequence of similar and dissimilar tasks. To our knowledge, no existing continual learning method has all these capabilities. This paper proposed a novel architecture CAT to achieve all these goals. Experimental results showed that CAT outperforms strong baselines. Our future work will focus on improving the accuracy of learning similar tasks (e.g., by considering task similarity computation as a regression problem rather than a binary classification problem), and improving the efficiency (e.g., by removing explicit similarity computation). We also plan to explore ways to use fewer labeled data in training.

## Acknowledgments

This work was supported in part by two grants from National Science Foundation: IIS-1910424 and IIS-1838770, a DARPA Contract HR001120C0023, and a research gift from Northrop Grumman.

## Broader Impact

An intelligent agent typically needs to learn many skills or tasks. Some of the tasks are similar to each other and some are distinct. It is desirable that the agent can learn these tasks without interference with each other and also improve its learning when there is shared/transferable knowledge learned in the past. As more and more chatbots, intelligent personal assistants and physical robots appear in our lives, we believe that this research will become more and more important. We could not see anyone will be put at disadvantage by this research. The consequence of failure of the system is that the system makes some incorrect classifications. Our task and method do not leverage biases in the data.

## Footnotes

[3]For example, one task is to classify fish and non-fish, and another task is to classify different kinds of fishes. Without knowing the user's specific task (task ID) at hand, the system will not know which is the right answer.

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
