[Supplementary Material]

# Supplementary Materials for
# Continual Learning of a Mixed Sequence of Similar and Dissimilar Tasks

**Zixuan Ke**[1], **Bing Liu**[1,*] and **Xingchang Huang**[2]
[1] Department of Computer Science, University of Illinois at Chicago
[2] ETH Zurich
{zke4, liub}@uic.edu, huangxch3@gmail.com

## 1   Detailed Performance

### 1.1   Detailed Results

From Table 1 in the main paper, we already knew the average results of each mixed sequence dataset and concluded that CAT is performing very well on both the overall average and the similar tasks while doing reasonable well for the dissimilar task datasets. It is also interesting to know the detailed performance of each random sequence in each mixed sequence dataset. We give these results in Table 1. The first column in the table shows the sequence id of each task sequence in each mixed sequence dataset. Each row gives the results of all models for the sequence, including the overall accuracy and the accuracy values for tasks in the dissimilar task dataset and for tasks in the similar task dataset. By comparing the rows for "EMNIST-10/20" and "CIFAR100-10/20", we can see how each model in the dissimilar task dataset are performing. Similarly, by comparing rows for "F-EMNIST" and "F-CelebA", we can see the performance on similar task datasets. More importantly, by comparing the rows for "Overall", we can see the average performance of all 20/30 tasks.

Comparing CAT and HAT, we can see that CAT clearly outperforms HAT for similar task datasets and is on par with, or close to HAT for dissimilar task datasets. This is expected because CAT is designed to not only avoid forgetting but also encourage knowledge transfer, regardless of the mixture of the task sequence. HAT, however, only deals with forgetting and has no mechanism to exploit the shared knowledge.

Regarding the other baselines, for M(EMNIST-10, F-EMNIST) and M(EMNIST-20, F-EMNIST), CAT consistently gives the best results for overall and similar task datasets. This indicates that CAT does very well in knowledge sharing. For dissimilar task datasets, while CAT is worse than ONE (which learns each task using an independent network and thus has no forgetting), it gives the best results among all others. This indicates CAT does well in avoiding forgetting.

For M(CIFAR100-10, F-CelebA) and M(CIFAR100-20, F-CelebA), ONE gives the best results for dissimilar task datasets, which is also not surprising. For similar task datasets, CAT is the best consistently. CAT works the best overall considering all baselines (see Table 1).

### 1.2   Execution Time and Number of Parameters

Table 2 reports CAT's execution times of training and testing. The training time is computed as the average training time per task. The testing time is computed as the total testing time for all tasks after all tasks are learned sequentially. We can see that our system CAT is highly efficient. Our experiments were run on GeForce GTX 1080 Ti with 10G GPU memory.

| Sequence | Average | NCL | ONE | EWC | UCL | HYPER | HYPER-R | PRO | PathNet | RPSNet | HAT | **CAT** |
|---|---|---|---|---|---|---|---|---|---|---|---|---|
| | | | | | | M(EMNIST-10, F-EMNIST) | | | | | | |
| 0 | Overall | 0.7319 | 0.7405 | 0.7330 | 0.7366 | 0.6181 | 0.4767 | 0.5530 | 0.5721 | 0.7102 | 0.7229 | **0.7559** |
| | EMNIST-10 | 0.9120 | **0.9448** | 0.9174 | 0.9216 | 0.8370 | 0.6951 | 0.9272 | 0.9157 | 0.8980 | 0.9351 | 0.9333 |
| | F-EMNIST | 0.5518 | 0.5363 | 0.5486 | 0.5517 | 0.3992 | 0.2583 | 0.1787 | 0.2285 | 0.5225 | 0.5106 | **0.5786** |
| 1 | Overall | 0.7416 | 0.7337 | 0.7445 | 0.7183 | 0.6491 | 0.5353 | 0.5296 | 0.6125 | 0.6877 | 0.7454 | **0.7840** |
| | EMNIST-10 | 0.9059 | **0.9430** | 0.9155 | 0.9114 | 0.8516 | 0.6959 | 0.9254 | 0.9329 | 0.8954 | 0.9392 | 0.9308 |
| | F-EMNIST | 0.5773 | 0.5243 | 0.5735 | 0.5253 | 0.4467 | 0.3747 | 0.1337 | 0.2921 | 0.4799 | 0.5516 | **0.6373** |
| 2 | Overall | 0.7225 | 0.7349 | 0.7272 | 0.7267 | 0.6201 | 0.4440 | 0.5488 | 0.5470 | 0.7137 | 0.7319 | **0.7754** |
| | EMNIST-10 | 0.9201 | **0.9457** | 0.9128 | 0.9122 | 0.8290 | 0.6906 | 0.9360 | 0.9121 | 0.9024 | 0.9341 | 0.9210 |
| | F-EMNIST | 0.5248 | 0.5241 | 0.5416 | 0.5411 | 0.4113 | 0.1974 | 0.1616 | 0.1819 | 0.5249 | 0.5297 | **0.6297** |
| 3 | Overall | 0.7371 | 0.7296 | 0.7377 | 0.7190 | 0.6224 | 0.5114 | 0.5272 | 0.6392 | 0.7092 | 0.7381 | **0.7698** |
| | EMNIST-10 | 0.9086 | **0.9471** | 0.9234 | 0.9203 | 0.8136 | 0.7685 | 0.9300 | 0.8953 | 0.8887 | 0.9259 | 0.9226 |
| | F-EMNIST | 0.5656 | 0.5121 | 0.5519 | 0.5177 | 0.4313 | 0.2543 | 0.1245 | 0.3830 | 0.5297 | 0.5503 | **0.6169** |
| 4 | Overall | 0.7134 | 0.7300 | 0.7272 | 0.7303 | 0.6255 | 0.4771 | 0.5367 | 0.5796 | 0.7015 | 0.7130 | **0.7700** |
| | EMNIST-10 | 0.9313 | **0.9377** | 0.9096 | 0.9152 | 0.8336 | 0.7767 | 0.9259 | 0.9257 | 0.8880 | 0.9340 | 0.9357 |
| | F-EMNIST | 0.4955 | 0.5222 | 0.5448 | 0.5453 | 0.4175 | 0.1775 | 0.1474 | 0.2335 | 0.5149 | 0.4920 | **0.6044** |
| | | | | | | M(EMNIST-20, F-EMNIST) | | | | | | |
| 0 | Overall | 0.8116 | 0.8277 | 0.8095 | 0.8199 | 0.7469 | 0.5352 | 0.6802 | 0.6842 | 0.7721 | 0.8108 | **0.8406** |
| | EMNIST-20 | 0.9251 | **0.9721** | 0.9334 | 0.9573 | 0.9098 | 0.6907 | 0.9691 | 0.9487 | 0.9178 | 0.9710 | 0.9605 |
| | F-EMNIST | 0.5844 | 0.5390 | 0.5618 | 0.5449 | 0.4209 | 0.2243 | 0.1024 | 0.1553 | 0.4806 | 0.4903 | **0.6009** |
| 1 | Overall | 0.8179 | 0.8267 | 0.8171 | 0.8168 | 0.7235 | 0.5665 | 0.6805 | 0.7301 | 0.7662 | 0.8183 | **0.8495** |
| | EMNIST-20 | 0.9457 | **0.9717** | 0.9316 | 0.9574 | 0.8870 | 0.8069 | 0.9688 | 0.9283 | 0.7702 | 0.9664 | 0.9583 |
| | F-EMNIST | 0.5622 | 0.5368 | 0.5880 | 0.5354 | 0.3964 | 0.0857 | 0.1038 | 0.3596 | 0.7758 | 0.5222 | **0.6320** |
| 2 | Overall | 0.8188 | 0.8245 | 0.8220 | 0.8096 | 0.7221 | 0.6303 | 0.6736 | 0.6799 | 0.7637 | 0.8224 | **0.8373** |
| | EMNIST-20 | 0.9413 | **0.9717** | 0.9455 | 0.9583 | 0.8926 | 0.8146 | 0.9587 | 0.9501 | 0.9086 | 0.9659 | 0.9498 |
| | F-EMNIST | 0.5739 | 0.5301 | 0.5751 | 0.5121 | 0.3813 | 0.2618 | 0.1035 | 0.2313 | 0.3699 | 0.5353 | **0.6123** |
| 3 | Overall | 0.7838 | 0.8175 | 0.8305 | 0.8215 | 0.7438 | 0.6654 | 0.6799 | 0.7403 | 0.7423 | 0.8098 | **0.8408** |
| | EMNIST-20 | 0.9155 | **0.9682** | 0.9448 | 0.9527 | 0.8994 | 0.7907 | 0.9652 | 0.9555 | 0.9110 | 0.9651 | 0.9613 |
| | F-EMNIST | 0.5205 | 0.5161 | 0.6021 | 0.5590 | 0.4328 | 0.4147 | 0.1095 | 0.2290 | 0.2434 | 0.4991 | **0.5997** |
| 4 | Overall | 0.7800 | 0.8260 | 0.8275 | 0.8254 | 0.7297 | 0.6488 | 0.6827 | 0.6944 | 0.7606 | 0.8207 | **0.8514** |
| | EMNIST-20 | 0.9076 | **0.9723** | 0.9411 | 0.9577 | 0.8962 | 0.8251 | 0.9681 | 0.9532 | 0.9229 | 0.9705 | 0.9530 |
| | F-EMNIST | 0.5246 | 0.5333 | 0.6004 | 0.5608 | 0.3966 | 0.2962 | 0.1118 | 0.2262 | 0.4946 | 0.5212 | **0.6483** |
| | | | | | | M(CIFAR100-10, F-CelebA) | | | | | | |
| 0 | Overall | 0.5475 | 0.6038 | 0.5908 | 0.5157 | 0.5434 | 0.3475 | 0.5892 | 0.5342 | 0.4868 | 0.5696 | **0.6138** |
| | CIFAR100-10 | 0.504 | **0.5895** | 0.5362 | 0.5314 | 0.4686 | 0.2041 | 0.5602 | 0.5503 | 0.4555 | 0.5755 | 0.564 |
| | F-CelebA | 0.5909 | 0.6182 | 0.6455 | 0.5000 | 0.6182 | 0.4909 | 0.6182 | 0.5182 | 0.5182 | 0.5636 | **0.6636** |
| 1 | Overall | 0.5240 | 0.5832 | 0.5969 | 0.5325 | 0.5401 | 0.3861 | 0.5745 | 0.5512 | 0.5254 | 0.5875 | **0.6270** |
| | CIFAR100-10 | 0.5117 | **0.5846** | 0.5302 | 0.5377 | 0.4621 | 0.1812 | 0.5580 | 0.5115 | 0.4871 | 0.5478 | 0.5268 |
| | F-CelebA | 0.5364 | 0.5818 | 0.6636 | 0.5273 | 0.6182 | 0.5909 | 0.5909 | 0.5909 | 0.5636 | 0.6273 | **0.7273** |
| 2 | Overall | 0.5499 | **0.6019** | 0.5854 | 0.5470 | 0.5182 | 0.4103 | 0.5882 | 0.5420 | 0.4946 | 0.5523 | 0.5948 |
| | CIFAR100-10 | 0.527 | **0.5857** | 0.5527 | 0.5395 | 0.4636 | 0.2478 | 0.5583 | 0.4930 | 0.4255 | 0.5592 | 0.5441 |
| | F-CelebA | 0.5727 | 0.6182 | 0.6182 | 0.5545 | 0.5727 | 0.5727 | 0.6182 | 0.5909 | 0.5636 | 0.5455 | **0.6455** |
| 3 | Overall | 0.5686 | 0.5969 | 0.6016 | 0.6015 | 0.5283 | 0.3282 | 0.5933 | 0.5504 | 0.4527 | 0.5804 | **0.6223** |
| | CIFAR100-10 | 0.5008 | **0.5847** | 0.5214 | 0.5393 | 0.4656 | 0.2019 | 0.5594 | 0.5280 | 0.3236 | 0.5789 | 0.5537 |
| | F-CelebA | 0.6364 | 0.6091 | 0.6818 | 0.6636 | 0.5909 | 0.4545 | 0.6273 | 0.5727 | 0.5818 | 0.5818 | **0.6909** |
| 4 | Overall | 0.5774 | 0.5974 | 0.5978 | 0.5647 | 0.5458 | 0.3793 | 0.5863 | 0.5740 | 0.4408 | 0.5513 | **0.6391** |
| | CIFAR100-10 | 0.5185 | **0.5858** | 0.5320 | 0.5385 | 0.4735 | 0.2131 | 0.5635 | 0.5390 | 0.3362 | 0.5845 | 0.5510 |
| | F-CelebA | 0.6364 | 0.6091 | 0.6636 | 0.5909 | 0.6182 | 0.5455 | 0.6091 | 0.6091 | 0.5455 | 0.5182 | **0.7273** |
| | | | | | | M(CIFAR100-20, F-CelebA) | | | | | | |
| 0 | Overall | 0.6039 | 0.6802 | 0.6215 | 0.6615 | 0.5982 | 0.4054 | 0.6778 | 0.6052 | 0.5225 | 0.6724 | **0.6994** |
| | CIFAR100-20 | 0.5922 | **0.7067** | 0.6186 | 0.6741 | 0.6110 | 0.3444 | 0.6894 | 0.6124 | 0.5383 | 0.6949 | 0.6764 |
| | F-CelebA | 0.6273 | 0.6273 | 0.6273 | 0.6364 | 0.5727 | 0.5273 | 0.6545 | 0.5909 | 0.4909 | 0.6273 | **0.7455** |
| 1 | Overall | 0.6032 | 0.6713 | 0.6447 | 0.6222 | 0.5963 | 0.3944 | 0.6758 | 0.6205 | 0.5449 | 0.6509 | **0.6932** |
| | CIFAR100-20 | 0.632 | **0.7069** | 0.6534 | 0.6696 | 0.6035 | 0.3325 | 0.6909 | 0.6262 | 0.5401 | 0.6855 | 0.6670 |
| | F-CelebA | 0.5455 | 0.6000 | 0.6273 | 0.5273 | 0.5818 | 0.5182 | 0.6455 | 0.6091 | 0.5545 | 0.5818 | **0.7455** |
| 2 | Overall | 0.6168 | 0.6739 | 0.6422 | 0.6091 | 0.6020 | 0.3744 | 0.6504 | 0.6324 | 0.5271 | 0.6596 | **0.6789** |
| | CIFAR100-20 | 0.6207 | **0.7063** | 0.6269 | 0.6682 | 0.6030 | 0.3162 | 0.6893 | 0.6349 | 0.5497 | 0.6939 | 0.682 |
| | F-CelebA | 0.6091 | 0.6091 | **0.6727** | 0.4909 | 0.6000 | 0.4909 | 0.5727 | 0.6273 | 0.4818 | 0.5909 | **0.6727** |
| 3 | Overall | 0.6099 | **0.6905** | 0.6246 | 0.6493 | 0.5717 | 0.4222 | 0.6564 | 0.6321 | 0.5524 | 0.6387 | 0.6760 |
| | CIFAR100-20 | 0.6103 | **0.7039** | 0.6323 | 0.6648 | 0.6030 | 0.3469 | 0.6892 | 0.6299 | 0.5604 | 0.6808 | 0.6777 |
| | F-CelebA | 0.6091 | 0.6636 | 0.6091 | 0.6182 | 0.5091 | 0.5727 | 0.5909 | 0.6364 | 0.5364 | 0.5545 | **0.6727** |
| 4 | Overall | 0.5752 | **0.6821** | 0.6133 | 0.6420 | 0.5707 | 0.3496 | 0.6807 | 0.5944 | 0.5584 | 0.6458 | 0.6742 |
| | CIFAR100-20 | 0.6128 | **0.7050** | 0.6427 | 0.6676 | 0.6061 | 0.2972 | 0.6893 | 0.5780 | 0.5648 | 0.6459 | 0.6386 |
| | F-CelebA | 0.5000 | 0.6364 | 0.5545 | 0.5909 | 0.5000 | 0.4545 | 0.6636 | 0.6273 | 0.5455 | 0.6455 | **0.7455** |

Table 1: Accuracy results of different models on the four mixed sequence datasets. Numbers in bold in each row are the best results.

| Dataset | Training time (h) | Testing time (s) |
|---|---|---|
| M(EMNIST-10, F-EMNIST) | 0.03 | 0.25 |
| M(EMNIST-20, F-EMNIST) | 0.05 | 0.43 |
| M(CIFAR100-10, F-CelebA) | 0.1 | 1.8 |
| M(CIFAR100-20, F-CelebA) | 0.15 | 3 |

Table 2: Execution time of CAT.

| Component | # Parameters (M) |
|---|---|
| M(EMNIST-10, F-EMNIST) | |
| Knowledge Base (input layer) | 1.5700 |
| Knowledge Base (hidden layer) | 4.0020 |
| Knowledge Transfer Attention (KTA) | 24.0160 |
| Task ID Embeddings (2 for $f_{mask}$ and 1 for $f_{KTA}$) | 0.1200 |
| Heads (20 for $f_{mask}$ and 20 for $f_{KTA}$) | 2.6693 |
| Transfer Network | 7.3649 |
| Reference Network | 3.6824 |
| Total | 43.4246 |
| M(EMNIST-20, F-EMNIST) | |
| Knowledge Base (input layer) | 1.5700 |
| Knowledge Base (hidden layer) | 4.0020 |
| Knowledge Transfer Attention (KTA) | 24.0160 |
| Task ID Embeddings (2 for $f_{mask}$ and 1 for $f_{KTA}$) | 0.1800 |
| Heads (30 for $f_{mask}$ and 30 for $f_{KTA}$) | 2.6693 |
| Transfer Network | 7.3649 |
| Reference Network | 3.6824 |
| Total | 43.4846 |
| M(CIFAR100-10, F-CelebA) | |
| Knowledge Base (input layer) | 6.1460 |
| Knowledge Base (hidden layer) | 4.0020 |
| Knowledge Transfer Attention (KTA) | 24.0160 |
| Task ID Embeddings (2 for $f_{mask}$ and 1 for $f_{KTA}$) | 0.1200 |
| Heads (20 for $f_{mask}$ and 20 for $f_{KTA}$) | 0.4802 |
| Transfer Network | 10.4049 |
| Reference Network | 5.2025 |
| Total | 50.3716 |
| M(CIFAR100-20, F-CelebA) | |
| Knowledge Base (input layer) | 6.1460 |
| Knowledge Base (hidden layer) | 4.0020 |
| Knowledge Transfer Attention (KTA) | 24.0160 |
| Task ID Embeddings (2 for $f_{mask}$ and 1 for $f_{KTA}$) | 0.1800 |
| Heads (30 for $f_{mask}$ and 30 for $f_{KTA}$) | 0.4802 |
| Transfer Network | 10.4049 |
| Reference Network | 5.2025 |
| Total | 50.4316 |

Table 3: Network size (number of parameters, regardless of trainable or un-trainable) of each component in CAT. Recall that the reference network has the same architecture as the transfer network, but half of its size.

Table 3 reports the number of parameters of different components of CAT. Note that different numbers of tasks lead to different sizes of the task ID embedding layers, and different input data sizes lead to different sizes of the input layer. Therefore, the total numbers of parameters are different for the four different mixed sequence datasets. Also note that as the number of tasks increases, the number of heads also increases, but this does not change the number of parameters because the number of classes per head decreases accordingly.

## Footnotes

*Corresponding author. The work was done when B. Liu was at Peking University on leave of absence from University of Illinois at Chicago.