[Reviews · NeurIPS 2020]

Review 1

Summary and Contributions: This paper proposes to leverage the information of task similarities to aid continual learning. It tries to transfer knowledge of similar tasks and prevent changing of important parameters of dissimilar tasks.

Strengths: The ideas of using the information of task similarities is interesting. The authors propose an intuitive and simple architecture, including the knowledge transfer module for similar tasks and a mask module to identify the important parameters (adopted from HAT) for dissimilar tasks.

Weaknesses: 1. From the perspective of HAT, the authors added a knowledge transfer module to learn from old similar tasks. However, this is at the expense of much more calculations, such as the knowledge transfer module (sec 3.2), the transfer model (L193), and the reference model (L198). There is no experiments showing the running time compared with HAT. Is it worthwhile to involve the extra calculations? 2. The HAT masked the important parameters using an attention module. This has identified similar tasks. Why do we need an extra module to do this? Is the motivation that discriminating similar and dissimilar tasks necessary? Similarity is a regression problem but not a classification problem. Why do you use a similar classifier (9) to do this?

Correctness: Yes.

Clarity: Yes.

Relation to Prior Work: Not really. More analysis to compare with HAT is needed, as shown in "weakness".

Reproducibility: Yes

Additional Feedback: See above. ------------- My initial major concern is the running time comparison. The authors have admitted that efficiency is a potential problem. They mentioned in the rebuttal that they use 10x training time to gain 3-5% higher accuracy. I am neutral to this now. Another concern is that I want to know the motivation to identify the similar or dissimilar tasks, as HAT has been able to mask the important parameters using an attention module. Is the motivation that discriminating similar and dissimilar tasks necessary? The authors cannot properly address this question in the rebuttal. My other concerns are answered properly. Therefore, I will change my score from "4: An okay submission, but not good enough; a reject." to "5: Marginally below the acceptance threshold."


Review 2

Summary and Contributions: This paper proposes a method to learn a sequence of mixed tasks in continual learning. For dissimilar tasks, the method deals with forgetting by using task masks, while for similar tasks, it uses knowledge transfer attention to selectively transfer the useful knowledge.

Strengths: The method proposed in this paper is novel and is relevant to the NeurIPS community. The claims in the paper are sound.

Weaknesses: I think the main weakness of the paper is the relatively simple network architecture used in the experiments (2-layer fully connected network), especially for CIFAR and CelebA data sets. For these data sets, a convolutional neural network (CNN) would be a more appropriate model. Can the proposed method be used with CNNs? Another weakness in my opinion is that the task similarity detection could be expensive. The method needs to train several f_{k -> t} models to detect similar tasks. It would be helpful if the paper could discuss more about the computational cost of the method. Besides, the space overhead when storing the masks should also be discussed.

Correctness: Yes.

Clarity: The paper is well written. But there are some minor points that are not very clear to me. 1. In Eq (1), m_l^(t) is the output of a sigmoid function. How did you make it into a binary value in Eq (2)? I do not think the paper explained this. 2. In Eq (5), N_t should be defined before the first use. 3. On L193, are the readout functions trained after applying the masks? 4. On L197, did you mean the "transfer network" instead of the "reference network"?

Relation to Prior Work: Yes.

Reproducibility: Yes

Additional Feedback: Post-rebuttal: I've read the other reviews and the rebuttal. The rebuttal has adequately addressed my concern regarding the CNN architecture with additional experimental results. I agree with other reviewers that running time is an issue for this method. I still think that this paper is marginally above the acceptance threshold due to the increase in accuracy.


Review 3

Summary and Contributions: The paper conjectures that the prior work on "task-level" continual learning either focuses on similar or dissimilar tasks. Consequently, authors consider a mix of both similar and dissimilar tasks during continual learning. The approach identifies tasks similar to the current one in order to update their parameters, while the dissimilar task parameters are fixed (using a learned mask over parameters). There were some concerns raised in the initial reviews, and the authors have provided a satisfactory response. I would suggest authors include the promised changes in the final version.

Strengths: * The introduction does a good job towards clearly defining the problem setting (i.e., incremental task learning instead of incremental classifier learning). * The main idea showcased in the paper is to identify a set of shared and masked parameters, which correspond to similar and dissimilar tasks respectively. In order to identify which tasks are similar, a validation dataset is used. During the update process, the gradients are blocked so that they dont affect the dissimilar task parameters and only update the similar task parameters to achieve backward knowledge transfer. * A knowledge base is used to summarize past task information and a task-specific attention is used to focus on the relevant information. * The paper employs some tricks to stablilize the learning process. One main trick is the use of annealing to stablitze the masks learned from embeddings.

Weaknesses: * The dataset choice seems arbitrary. Since authors are defining a new setting, they should elaborate why specifically FEMNIST and FCelebA are used to create similar and dissimilar pairs. * Relation to relevant prior work is not mentioned and elaborated. For example, Rajasegaran, et al. "Random path selection for continual learning." NeurIPS'19 also propose a similar masking based approach to learn non-overlapping paths for dissimilar tasks. Similarly, PathNet (Evolution Channels Gradient Descent in Super Neural Networks) selectively masks out irrelavent model paramters. These papers should be cited and disucssed (preferably compared against) in this manuscript. * To my understanding, the notion of similar and dissimilar tasks is not accurate. E.g., the prior works on task incremental learning have both sets of similar and dissimilar tasks. (E.g., consider CIFAR100 classes in GEM - NeurIPS'17). In fact the considered set of similar and dissimilar tasks is not too different from the ones considered in earlier works. Specifically, consider a seminal work from Li & Hoeim, "Learning without forgetting" (TPAMI), where different datasets such as ImageNet/Places365/VOC/CUB/Scenes/MNIST are considered in continual learning experiments). Nevertheless, the proposed splits and dataset choices should be properly motivated and the authors should also report some experiments on previously considred protocols for fair benchmarking against existing methods. * The annealing strategy is somewhat similar to controller proposed in iTAML (iTAML : An Incremental Task-Agnostic Meta-learning Approach - CVPR'20). * The approach assumes that the task ID is known beforehand. Although this is consistent with some prior works, isn't it a bit restrictive in practical settings? It would be good to explain some application scenarios where tasks ID can be known to motivate the readers. * Equation 3 is wrong, it should be explicitly written. * The caption of Figure 1 should have some description for the MTCL architecture (a) as well.

Correctness: The method generally seems correct. The empirical methododlogy is reasonable, however, I have do have some concerns as mentioned under weaknesses and relation to prior work sections.

Clarity: The paper is overall clearly written with nice visualizations and easy to follow structure.

Relation to Prior Work: This is the biggest concern for me is that relevant prior work is neither cited nor properly acknolwedged. Further, there should be comparisons with similar previous work. If the authors argue that those were for a different incremental learning settings, then they must implement a simple baseline version of previous models to clearly show the advanatage of proposed improvements. In addition, the authors devise a new protocol without much justification. I think it needs to be better motivated along with more extensive comparisons.

Reproducibility: Yes

Additional Feedback:


Review 4

Summary and Contributions: The paper proposes a novel model for learning a sequence of tasks, some of which are similar and other dissimilar, while previous work focused on only similar or dissimilar tasks. The proposed model learns embeddings for each tasks and predicts binary masks for each layer to gate the gradient. The network has a knowledge base network with a task classification loss, and a knowledge transfer attention network conditioned on all similar task's hidden representations from the former network, again with a classification loss. The model is compared to a set of benchmarks in several sets of classification tasks and outperforms all the benchmarks in terms of accuracy for mixed similar/dissimilar sequences. One of the benchmarks, ONE, tends to outperform the proposed model on sequences of only dissimilar tasks, but the proposed model again outperforms all benchmarks for sequences of only similar tasks.

Strengths: Novel model and setup evaluated against a number of recent benchmark models. Ablation study supports the models features.

Weaknesses: The tasks considered are on small datasets with small numbers of classes and the paper only considers classification tasks in experiments.

Correctness: The results seem to support the claims. It would be good to have a discussion/comparison to data and tasks used for evaluating models in previous work.

Clarity: Yes.

Relation to Prior Work: The paper has a good related work section.

Reproducibility: Yes

Additional Feedback: Comments: - Define x, y in the paper. - There seems to be a different f_mask and f_KTA for each tasks which is not clear from the notation or the figure.

[Author Response · NeurIPS 2020]

Thank you (R1, R2, R3, R4) for your insightful comments. We will release the code and fix the minor issues.

**Cost of Task Similarity Detection (R1,R2).** Yes, it incurs additional computation, but it is reasonable. MTCL's
training time (h) ranges from 0.03 to 0.15, while HAT ranges from 0.0035 to 0.0039. MTCL's number of parameters
(M) ranges from 43.4 to 50.4, while HAT ranges from 7 to 10.5. See Sec. 2.3 in *Appendix* for more details. As MTCL
makes large gains compared to strong baselines (at least 3%, up to 30%), we believe the performance gain is worth the
additional computation. Also importantly, the proposed problem is more general and has not been attempted before.

**Use of Validation Set (R1).** Validation set (part of the training set) has been used in many continual learning techniques,
e.g., HAT, EWC and ProgressiveNet, for parameter tuning. In our case, we also use it for similarity detection.

**Necessity of Similarity Detection (R1).** It may be possible to design an algorithm without explicit similarity detection.
In our current model, it is needed; otherwise, we cannot generate TSV (Task Similarity Vector). Ablation study (Table
3) shows that MTCL degrades severely without TSV (-*TSV*) as TSV allows only the units of similar tasks to be updated;
otherwise, severe forgetting occurs. HAT does not detect task similarity. It only detects which units are associated with
which tasks and block them when learning future tasks.

**Regression v.s. Classification (R1).** We regard task similarity detection as a binary classification problem. The
detection results (TSV) are used to determine what parameters to update (i.e. only those units associated with similar
tasks can be updated). It is also interesting to consider this as a regression problem. We can study that in the future.

**Simple Architecture - does CNN work for CIFAR and CelebA?**
**(R2).** As we want to make our experiments uniform, we adopt the
same backbone MLP architecture for experiments. Replacing MLP
with CNN is straightforward for MTCL. We did that (see CNN's overall
results in Table 1, better than the MLP results) - MTCL again outperforms
the baselines. ONE is one-task learning, learning each task independently.

| Overall | NCL | ONE | HAT | MTCL |
|---|---|---|---|---|
| M(CIFAR100-10,F-CelebA) | 0.6155 | 0.6764 | 0.6178 | **0.6831** |
| M(CIFAR100-20,F-CelebA) | 0.6931 | 0.7428 | 0.6946 | **0.7468** |

Table 1

| Overall | PathNet | RPSNet | MTCL |
|---|---|---|---|
| M(EMNIST-10,F-EMNIST) | 0.5901 | 0.7044 | **0.7710** |
| M(CIFAR100-10,F-CelebA) | 0.5504 | 0.4801 | **0.6194** |
| M(EMNIST-20,F-EMNIST) | 0.7049 | 0.7620 | **0.8439** |
| M(CIFAR100-20,F-CelebA) | 0.6169 | 0.5410 | **0.6843** |

Table 2

**Additional Baselines, Dataset Choice and How Similar Tasks Different**
**from Existing Systems' Setting (R3).** Our work is different from existing systems: (1) We consider a *mixed sequence*
*of tasks*. Forgetting mainly affects learning *dissimilar* tasks. For similar tasks, forgetting is not a major issue (see lines
274-277). Knowledge transfer is important. Regarding *dissimilar* tasks, we adopt datasets CIFAR100 and EMNIST
as they are commonly used benchmarks. They form *dissimilar* tasks as their classes have little knowledge sharing.
For *similar* tasks, we adopt the datasets in federated learning (see lines 4-14 in Appendix). Federated learning is to
train through model aggregation rather than the normal data aggregation and keep local data on the local device. These
datasets naturally consist of *similar* tasks. How to leverage the shared knowledge is important for similar tasks, (2)
Existing models mainly focus on addressing forgetting because they don't consider *mixed* datasets, but only dissimilar
datasets/tasks. Although some models do limited forward transfer, they still mainly deal with forgetting. They cannot
improve the results of similar tasks as we do. For example RPSNet (Rajasegaran et al., 2019) and PathNet (Fernando et
al., 2017) detect which path to reuse but do not allow the used parameters to be updated. These are insufficient because
similar tasks' parameters should be allowed to update for both forward and backward knowledge transfer. We believe
this is an important contribution of ours. LwF and GEM also use the previous model in learning the new task, but it is
also for dealing with forgetting only and it cannot improve similar task performance through forward and backward
transfer. **Additional comments**: (1) We have conducted new experiments on the two suggested baselines: PathNet and
RPSNet. Table 2 shows their overall accuracy results (average over 5 random sequences). Detailed results will be added
to the paper. MTCL outperforms PathNet and RPSNet considerably. (2) About motivation of our work, please refer to
lines 15-36. We will explain more in the revised paper. (3) Our dataset choice: (a) datasets in federated learning give us
similar tasks, and (b) the similar datasets should be paired with our dissimilar datasets EMNIST and CIFAR in terms of
the image size to copmose the mixed task sequences. Other public datasets (Caldas et al., 2018) in federated learning
are either text or with different image/input sizes, which are unsuitable for the data pairing.

**Other Comments (R1,R2,R3,R4).** *(1) How to make the output of the sigmoid function binary.* It is binaried by adding
the hyper-parameter $s$ in Eq. 1. $s$ is annealed (see lines 139-144). When $s$ is very large ($s \to \infty$), the output of sigmoid
$m_l^{(T)} \to \{0, 1\}$, approximating to a unit step function. *(2) Are the readout functions trained after applying the masks?*
Yes. We first use the saved mask $\{m_l^{(k)}\}$ to get the representation/features for $x^{(t)}$ produced by the task $k$ model in
the KB and then train the readout function. *(3) The annealing strategy is somewhat similar to controller proposed in*
*iTAML.* Yes, but one notable difference is that the controller in iTAML is to balance between plasticity and stability,
while ours is to train a binary mask. We will cite and compare with iTAML in the revised paper. *(4) Task ID has to be*
*known.* Task ID is used in all task-based continual learning algorithms. This is so because the tasks can be completely
different or overlapping and thus need separate sub-models in the neural network. For example, one task is to classify
fish and non-fish, and another task is to classify different kinds of fishes. Without knowing the user's specific task at
hand, the system will not know which is the right answer.

[Meta-Review · NeurIPS 2020]

The reviews on this paper are mixed; even reviewers supporting acceptance stated during the discussion phase that this paper would benefit from another round of revisions before publication. The paper explores a combination of several ideas, most notably the separation of similar and dissimilar tasks along with masking of critical parameters, with significant improvements during continual learning. The results seem solid and demonstrate the strength of the proposed method. This is a nice contribution. However, the reviewers had significant concerns about the computational complexity of the proposed method. There are also a variety of clarity issues throughout the paper, including both confusing presentation or wording, and minor misstatements about previous work — the paper must have these corrected before publication. In particular, there is one aspect of the review that absolutely MUST be corrected before publication. This area chair agrees with the reviewer 3’s concerns that the paper includes misstatements regarding previous work, and disagrees with the authors’ rebuttal to this point. Many previous works did explicitly consider mixes of similar and dissimilar tasks, such as in the references mentioned in the reviews and those cited in [Chen & Liu, Lifelong Machine Learning, 2nd Edition, Ch 9.3.5]. This makes the author's statement "To the best of our knowledge, no existing work has been done to learn a sequence of mixed similar and dissimilar tasks that deal with forgetting and improve learning at the same time" in Line 93 (now) inaccurate. This authors have a responsibility to correct this before publication. Note that these earlier methods would likely not have the same level of performance as this method. Regardless, the technique used in this paper is novel and has shown good performance gains, so it would be beneficial to the community. The private comments to the Area Chair were taken into consideration. Minor notes: -Citations should all be in [] or () so that they’re separated from the main text. -There is a mistake in line 86. Backward transfer was also investigated in some of the works mentioned on line 85 under the name "reverse transfer", not just in Wang et al 2019. -In response to the second concern you mentioned privately to the Area Chair, the issue likely stems from that conclusion not being mentioned explicitly in lines 286-290, and only coming from careful analysis of Table 3. To prevent such a misunderstanding, you should explicitly state what the ablative study shows in terms of the importance of similarity detection.